# Biallelic variants in *MAD2L1BP* (*p31*comet) cause female infertility characterized by oocyte maturation arrest

Lingli Huang[1,2†], Wenqing Li[1,3†], Xingxing Dai[4,5†], Shuai Zhao[6†], Bo Xu[1†], Fengsong Wang[7], Ren-Tao Jin[1], Lihua Luo[1], Limin Wu[1], Xue Jiang[1,3], Yu Cheng[1,3], Jiaqi Zou[1,3], Caoling Xu[1,3], Xianhong Tong[1], Heng-Yu Fan[4]*, Han Zhao[6]*, Jianqiang Bao[1,3]*

[1]Reproductive and Genetic Hospital, The First Affiliated Hospital of USTC, Division of Life Sciences and Medicine, University of Science and Technology of China, Hefei, China; [2]Center for Reproductive Medicine, Department of Obstetrics and Gynecology, Anhui Provincial Hospital Affiliated to Anhui Medical University, Hefei, China; [3]Hefei National Laboratory for Physical Sciences at Microscale, Biomedical Sciences and Health Laboratory of Anhui Province, University of Science and Technology of China (USTC), Hefei, China; [4]Life Sciences Institute, Zhejiang University, Hangzhou, China; [5]International Institutes of Medicine, the Fourth Affiliated Hospital of Zhejiang University School of Medicine, Yiwu, China; [6]Hospital for Reproductive Medicine, State Key Laboratory of Reproductive Medicine and Offspring Health, Key Laboratory of Reproductive Endocrinology of Ministry of Education, Shandong Key Laboratory of Reproductive Medicine, Shandong Provincial Clinical Research Center for Reproductive Health, Shandong University, Jinan, China; [7]School of Life Science, Anhui Medical University, Hefei, China

*For correspondence:
hyfan@zju.edu.cn (H-YuF);
hanzh80@sdu.edu.cn (HZ);
jqbao@ustc.edu.cn (JB)

†These authors contributed equally to this work

Competing interest: The authors declare that no competing interests exist.

**Abstract** Human oocyte maturation arrest represents one of the severe conditions for female patients with primary infertility. However, the genetic factors underlying this human disease remain largely unknown. The spindle assembly checkpoint (SAC) is an intricate surveillance mechanism that ensures accurate segregation of chromosomes throughout cell cycles. Once the kinetochores of chromosomes are correctly attached to bipolar spindles and the SAC is satisfied, the MAD2L1BP, best known as p31comet, binds mitosis arrest deficient 2 (MAD2) and recruits the AAA+-ATPase TRIP13 to disassemble the mitotic checkpoint complex (MCC), leading to the cell-cycle progression. In this study, by whole-exome sequencing (WES), we identified homozygous and compound heterozygous *MAD2L1BP* variants in three families with female patients diagnosed with primary infertility owing to oocyte metaphase I (MI) arrest. Functional studies revealed that the protein variants resulting from the C-terminal truncation of MAD2L1BP lost their binding ability to MAD2. cRNA microinjection of full-length or truncated *MAD2L1BP* uncovered their discordant roles in driving the extrusion of polar body 1 (PB1) in mouse oocytes. Furthermore, the patient's oocytes carrying the mutated *MAD2L1BP* resumed polar body extrusion (PBE) when rescued by microinjection of full-length *MAD2L1BP* cRNAs. Together, our studies identified and characterized novel biallelic variants in *MAD2L1BP* responsible for human oocyte maturation arrest at MI, and thus prompted new therapeutic avenues for curing female primary infertility.

## Editor's evaluation

This important study identifies three independent patient mutations in MAD2L1BP (p31 comet) that cause infertility. Consistent with the known functions of p31 comet, convincing experiments in mouse oocytes imply that infertility could be caused by a failure to silence the spindle assembly checkpoint, though the mechanism was not determined. Although the sample size is small, a rescue experiment in human oocytes promises the potential for therapy.

## Introduction

Infertility is increasingly becoming a global health issue that affects 10% to 15% (e.g., approximately 186 million) of couples at reproductive age around the world (*Inhorn and Patrizio, 2015*). Currently, many couples benefit from assisted reproductive techniques (ARTs), such as *in vitro fertilization* (IVF) or *intracytoplasmic sperm injection* (ICSI), to have their own babies. However, among the patients seeking ART treatment in the clinic, some of the females suffer from recurrent failure even through repeated IVF/ICSI attempts. These female patients with primary infertility were most often attributed to functional deficiency in the oocytes, such as oocyte maturation arrest, premature oocyte death, fertilization failure, or preimplantation embryonic arrest (*Sang et al., 2021*). Oocyte maturation is referred to as meiotic resumption of oocytes from the germinal vesicle (GV) to the metaphase II (MII) stage in vivo or in vitro, whereby the oocytes acquire the developmental competence. In the clinic, oocyte maturation arrest is characterized by the frequent occurrence of oocytes mostly or entirely arrested at the GV or metaphase I (MI) stage, representing a rare but highly challenging disease for treatment through traditional ART (*Mehlmann, 2005*; *Mrazek and Fulka, 2003*; *Beall et al., 2010*).

Unlike the male germline, female germ cells initiate meiosis very early during embryonic development once primordial germ cells transit and colonize the genital ridge in mice and humans (*Hilscher et al., 1974*; *McLaren, 2003*). Nonetheless, following the synapsis and recombination between homologous chromosomes, meiotic progression ceases in the embryonic gonad, and the oocytes remain arrested at the diplotene stage in meiotic prophase I during postnatal oocyte growth in growing follicles (*McLaren, 2003*). After birth, a limited number of primordial follicles are activated and recruited from the primordial follicle pool to develop sequentially into primary follicles, secondary follicles, and antral follicles. This process is concurrent with the oocyte cytoplasmic and meiotic maturation process whereby the oocytes continuously grow with increased size and acquire the ability to resume meiotic division upon luteinizing hormone surge (*Mehlmann, 2005*; *Park et al., 2004*). The fully grown oocyte is morphologically characterized by the appearance of GV, representative of the oocyte nucleus with a bulgy nucleolus that can be readily identified under phase-contrast microscopy. Morphologically, meiotic resumption is characterized by chromatin condensation and the dissolution of nuclear membranes, termed 'GV breakdown' (GVBD). This is followed by the cell-cycle progression to MI and subsequent extrusion of the first polar body (PB1), culminating in MII arrest, which is ready for fertilization (*Mihajlović and FitzHarris, 2018*; *Adhikari and Liu, 2014*). This prolonged meiotic prophase I arrest at diplotene stage in conjunction with the discontinuous, hormone-triggered progression to MII stage in oocytes is highly genetically regulated and spatiotemporally coordinated. In agreement with this conclusion, numerous genetically engineered mouse models (GEMMs) have uncovered the genetic factors that are required to drive accurate and timely meiotic progression. For instance, genetic ablation of *Marf1*, *Cdc25b*, or *Pde3a* led to GV arrest (*Su et al., 2012*; *Lincoln et al., 2002*; *Masciarelli et al., 2004*), while deletion of *Ccnb3*, *Cks2*, or *Mlh1* caused MI arrest in mouse oocytes (*Karasu et al., 2019*; *Spruck et al., 2003*; *Woods et al., 1999*). Occasionally, genetic abrogation can cause a mixed phenotype. For example, mouse oocytes deficient in *Mlh3* arrested at both the MI and MII stages (*Lipkin et al., 2002*), whereas *Ubb*-null oocytes ceased at both the GV and MI stages (*Ryu et al., 2008*). Although these mouse genetic models provide insightful mechanisms underlying oocyte meiosis, the causative deleterious variants responsible for oocyte maturation arrest resulting in human infertility remain scarce. Until recently, only a limited number of protein-coding variants have been discovered to account for human oocyte maturation arrest, including *TUBB8* (*Feng et al., 2016*) (MIM: 616768), *PATL2* (*Maddirevula et al., 2017*; *Chen et al., 2017*; *Huang et al., 2018*) (MIM:614661), and *TRIP13* (*Zhang et al., 2020*) (MIM: 604507). However, mutations in these genes can only account for a very small proportion (<30%) of patients with this syndrome, and the deleterious causative variants in most cases remain largely unknown (*Feng et al., 2016*; *Maddirevula*

**Table 1.** Clinical characteristics of affected individuals and their retrieved oocytes.

| Age (years) | Duration of infertility (years) | IVF/ICSI cycles | Protocol | Total number of oocytes retrieved | GV oocytes | MI oocytes | PB1 oocytes | Fertilized oocytes | Cleaved embryos |
|---|---|---|---|---|---|---|---|---|---|
| Family 1 (II-1) | | | | | | | | | |
| 22 | 2 | 1(IVF) | Modified ultra-long | 11 | 0 | 11 | 0 | 0 | 0 |
| | | 2(IVF) | Mild stimulation | 6 | 1 | 5 | 0 | 0 | 0 |
| | | 3(IVF) | PPOS | 13 | 0 | 13 | 0 | 0 | 0 |
| Family 2 (II-1) | | | | | | | | | |
| 31 | 6 | 1(IVF) | PPOS | 9 | 0 | 9 | 0 | 0 | 0 |
| Family 3 (II-1) | | | | | | | | | |
| 34 | 9 | 1(IVF) | Natural | 1 | 0 | 1 | 0 | 0 | 0 |
| | | 2(IVF) | Short | 4 | 0 | 4 | 0 | 0 | 0 |

GV, geminal vesicle; MI, metaphase I; PB1, first polar body; PPOS, progestin-primed ovarian stimulation.

*et al., 2017*; *Chen et al., 2017*; *Huang et al., 2018*; *Zhang et al., 2020*; *Christou-Kent et al., 2018*). Therefore, there is an urgent need to screen the causal factors among the female patients suffering from recurrent reproductive failure for better genetic diagnosis and treatment in the future.

MAD2(MAD2L1) is a key component of mitotic checkpoint complex (MCC). MAD2L1-binding protein (MAD2L1BP) (MIM: 618136), also called p31[comet], mimicking the structure of mitosis arrest deficient 2 (MAD2), physically interacts with MAD2 through a conformation-specific binding and it functions as an adapter between MAD2 and TRIP13 (*Yang et al., 2007*; *Alfieri et al., 2018*). In mitosis, timely disassembly of the MCC and hence the silencing of the spindle assembly checkpoint (SAC) rely on the joint action of both p31[comet] and TRIP13 (*Alfieri et al., 2018*). Whole-body p31[comet] knockout mice die soon after birth and have reduced hepatic glycogen (*Choi et al., 2016*). In rice, P31[comet] plays an important role in meiosis. The P31[comet] mutant rice was normal in vegetative growth but showed complete sterility (*Ji et al., 2016*). However, whether the p31[comet] protein plays any role in meiosis in mammals remains unknown. In this study, we identified homozygous and compound heterozygous *MAD2L1BP* variants in three families with female patients diagnosed with primary infertility owing to oocyte MI arrest. Our findings suggest that biallelic *MAD2L1BP* variants contribute to oocyte MI arrest in women with primary infertility, and p31[comet] is required for human oocyte maturation.

## Results

### Identification of biallelic variants in *MAD2L1BP* underlying human oocyte maturation arrest at the MI stage

A cohort of 50 female patients with primary infertility displaying the oocyte maturation arrest at the MI stage was recruited from the Reproductive Center of the First Affiliated Hospital of USTC (University of Science and Technology of China) and the Reproductive Hospital at Shandong University between July 2014 and October 2021. All the patients had normal karyotypes (46, XX) and had undergone at least one cycle of IVF or ICSI treatment, and IVF or ICSI treatments of the three patients with variants in *MAD2L1BP* are shown (*Table 1*). The oocytes and blood samples were donated by individuals who had provided the written, informed consent paperwork. The DNA samples were subjected to conventional whole-exome sequencing (WES) screening in order to underpin the genetic variants. We screened the candidate variants based on the following criteria: (a) rare variants with a minor allele frequency (MAF) below 1% in five public databases: 1000 genome, dbSNP, gnomAD, EVS, and Exome Aggregation Consortium (ExAC); (b) nonsynonymous exonic or splice site variants, or frame-shift INDEL; (c) heterozygous variant that is also carried by the parents; (d) known RNA expression in

our *in-house* oocyte expression database; (e) homozygous variants were prioritized in consanguineous families (*Figure 1A*, see also Materials and methods). This pipeline analyses led us to narrow down the candidate variants to a short list in three families (12, 14, and 11 candidate genes in the respective families) (*Table 2*, *Table 2—source data 1*), and eventually to focus on one shared gene variant, namely *MAD2L1BP*, wherein two homozygous nonsense variants (c.853C>T [p.R285*] and c.541C>T [p.R181*]) of *MAD2L1BP* from two unrelated families (Family 1 and Family 3), respectively, and two compound heterozygous variants (a frameshift mutation p.F173Sfs4* and an intronic mutation c.21-94G>A) of *MAD2L1BP* in Family 2 (*Table 2—source data 1*, *Table 3*).

In the first case, the female patient (F1: II-1) was from a consanguineous family and underwent three unsuccessful cycles of IVF with 29 out of a total of 30 superovulated oocytes arrested at the MI stage (*Figure 1A*, *Figure 1C*, *Table 1*). WES analysis following a set of bioinformatic pipelines identified a homozygous nonsense mutation in NM_001003690 (*MAD2L1BP*): c.853C>T [p.R285*] (*Figure 1A and B*). Her younger brother (F1: II-2), 22 years of age, carried the same homozygous mutation (*Figure 1A*, *Figure 1—figure supplement 2*). Further Sanger sequencing after PCR amplification confirmed that their consanguineous parents were both heterozygous for this mutation. The second patient was diagnosed with primary infertility with no history of consanguinity. She had gone through one cycle of IVF, and all nine collected oocytes were arrested at MI (F2: II-1) (*Figure 1C*, *Table 1*). A heterozygous frameshift mutation c.518delT [p.F173Sfs4*] in *MAD2L1BP* was uncovered by WES (*Figure 1A*,*Table 3*), which was inherited from her mother. Subsequent Sanger sequencing of ~200 bp intron-flanking sequences of *MAD2L1BP* validated a heterozygous intronic variant c.21–94G>A in the patient (F2: II-1), which was transmitted from her father (*Figure 1A–C*, *Table 3*). In the third family, the female patient had a total of five superovulated oocytes all arrested at MI after two unsuccessful cycles of IVF. WES and subsequent PCR sequencing revealed a homozygous mutation c.541C>T [p.R181*] in *MAD2L1BP* (*Figure 1A and B*, *Table 1*). This mutation was inherited from her parents respectively and likely resulted in the production of a much shorter C-terminally truncated MAD2L1BP protein compared with p.R285* variant from the first patient (F1: II-1).

The nonsense variants p.R285* and p.R181* occurred at low allele frequencies (3/248598 and 2/250850, respectively) in the gnomAD browser, and the frameshift mutation p.F173Sfs4* was not found in the database (*Table 3*). However, the intronic variant c.21-94G>A (rs142226267) had an allele frequency of 111/31250 and homozygote frequency of 2/31250 in the gnomAD browser, which was initially filtered out through the WES analysis pipeline but subsequently reconsidered for further evaluation of its pathogenicity (*Table 3*). The nonsense variants p.R285* and p.R181* and the frameshift mutation p.F173Sfs4* gave rise to a premature termination codon (PTC) in the resultant *MAD2L1BP* mRNA. For the intronic variant c.21-94G>A, it was predicted to be a cryptic splicing acceptor site of exon 2 by NNsplice and ASSP, presumably leading to aberrant splicing of Exon 2 with additional intronic sequences resulting in a PTC introduction (*Table 3*).

## C-terminal truncation of MAD2L1BP abolished its interaction with MAD2, a key kinetochore protein

MAD2L1BP is a relatively small-sized protein consisting of only 306 amino acids, which are highly evolutionarily conserved across metazoan species (*Figure 1—figure supplement 1* and *Figure 2— figure supplement 1*). Interestingly, there is a shorter isoform that differs exclusively at the N-terminus compared to the longer isoform in humans (*Figure 1—figure supplement 1*). By analyzing the published RNA-seq databases, we found that *MAD2L1BP* mRNAs are highly abundant in both human and mouse oocytes, especially those at advanced stages (*Figure 2—figure supplement 1*), indicative of an important role of MAD2L1BP in oocyte meiotic maturation in mammals. To determine whether those variants impact *MAD2L1BP* mRNA expression, we next designed quantitative PCR (qPCR) primers against *MAD2L1BP* mRNAs. We discovered that p.R285* mutation did not affect the expression levels of *MAD2L1BP* mRNAs in the first patient (F1: II-1) (*Figure 2A*), thus possibly producing a truncated MAD2L1BP protein loss of C-terminal 22 amino acids (*Figure 1B*). In the second patient carrying the compound heterozygous variants, the p.F173Sfs4* mutation would elicit frameshift reading causing nonsense-mediated mRNA decay or generate a much shorter C-terminally truncated MAD2L1BP protein. To investigate whether the c.21-94G>A (rs142226267) variant affects splicing of *MAD2L1BP* mRNA, we initially tried the cDNA amplification using total RNA extracted from the patient's peripheral blood sample. However, we failed to amplify the predicted bands after several

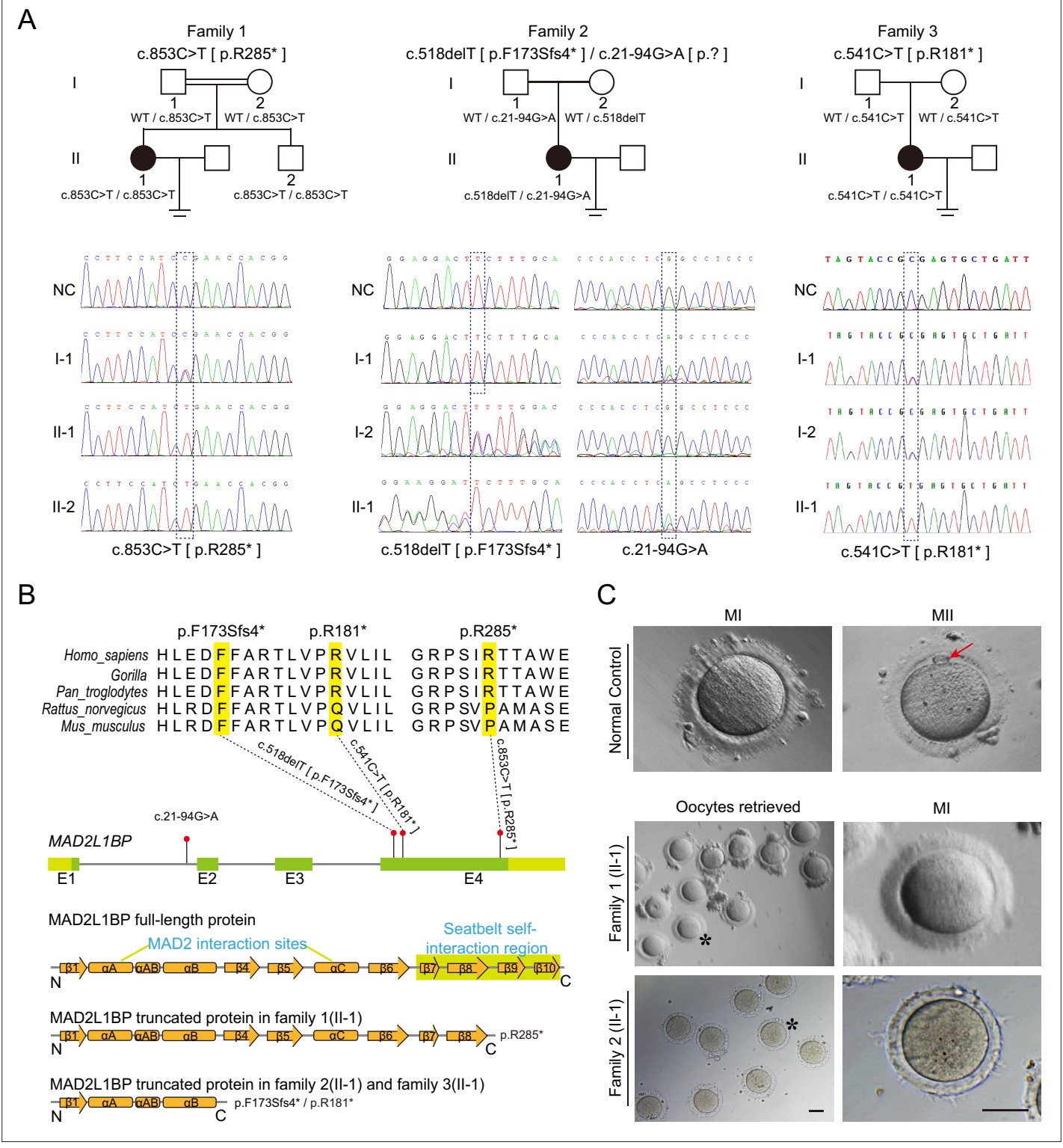

**Figure 1.** Identification of pathogenic variants in *MAD2L1BP* from three unrelated families. (**A**) Pedigrees of three female patients with oocyte maturation arrest at the MI stage from three unrelated families. The first case was from a consanguineous family, while the other two were not. The patients carried homozygous c.853C>T [p.R285*] and c.541C>T [p.R181*] variants from Family 1 and Family 3, respectively, while the patient from Family 2 had a compound heterozygous mutation c.518delT and c.21–94G>A as indicated. The chromatograms below show the Sanger sequencing results of the PCR-amplified fragments containing the respective variants in each family. (**B**) Phylogenetic conservation of the identified amino acid mutations in MAD2L1BP. The positions of all mutations are indicated in the schematic genomic structure of *MAD2L1BP*. The MAD2-interacting site and the seatbelt

*Figure 1 continued on next page*

*Figure 1 continued*

self-interaction region for MAD2L1BP are labeled in the secondary structure of MAD2L1BP protein at the bottom. All identified mutations residing in Exon 4 of *MAD2L1BP* generated the premature STOP codon, likely resulting in truncated proteins. (C) The representative morphology of normal and affected individual oocytes. The normal MI and MII oocytes are shown in the top panel. The red arrow points to the first polar body (PB1). Oocytes retrieved from Family 1 (II-1) and Family 2 (II-1) were arrested at the MI stage. The asterisks indicate the oocytes with a magnified view in the right panel. MI: metaphase I; MII: metaphase II. Scale bar, 80 μm.

The online version of this article includes the following figure supplement(s) for figure 1:

**Figure supplement 1.** Multi-sequence alignment of amino acids for MAD2L1BP (p31^comet) (A) and its interacting partner MAD2 (B) orthologs from eutherian species with Jalview.

**Figure supplement 2.** Semen analysis of the individual II-2 in Family 1 with a homozygous mutation (c.853C>T [p.R285*]) in *MAD2L1BP*.

rounds of attempts by optimization of the primer designs and PCR conditions, presumably owing to the extremely low abundance of aberrant *MAD2L1BP* mRNA in the blood (*data not shown*). Therefore, we next conducted an in vitro splicing assay by synthesizing a plasmid that comprised the Exon 1, Exon 2, and Exon 3 as well as the flanking intronic sequences in the *MAD2L1BP* gene. Not surprisingly, the aberrant splicing around Exon 2 was repeatedly observed by sequencing the transcribed spliced mRNA product after transfection into 293T cells in vitro (*Figure 2—figure supplement 2*). In support of this finding, the qPCR assay validated that the *MAD2L1BP* mRNA levels were significantly reduced in the blood from the second patient (*Figure 2B*). We were unable to quantitatively measure the effect of p.R181* mutation on *MAD2L1BP* mRNA expression in the third patient owing to the unavailability of blood samples. However, the p.R181* mutation would only be possible to generate a much shorter MAD2L1BP protein than p.R285* variant in the first patient (F1: II-1), leading to a complete loss of 126 amino acids at the C-terminus (*Figure 1B*).

Because the p.R285* variant likely generated the longest MAD2L1BP transcript isoform among other variants, resulting in the loss of C-terminal 22 residues for MAD2L1BP (Figure 2D), we thus next chose this variant as a representative to investigate whether the MAD2L1BP mutations found in patients would induce functional deficiency. MAD2L1BP has been shown to act as an adaptor that tightly binds MAD2 and recruits TRIP13 (*Alfieri et al., 2018*; *Brulotte et al., 2017*; *Ma and Poon, 2016*). At the protein level, MAD2L1BP harbors highly conserved central 'α-helix'- and flanked 'β-sheet'-organized domains, which structurally mimick MAD2 and physically interact with each other in vivo at the MAD2 dimerization interface (*Yang et al., 2007*; *Figures 1B and 2F*). In accordance with *MAD2L1BP*, the mRNA expression levels of *MAD2* were also high in both human and mouse oocytes (*Figure 2—figure supplement 1*). MAD2 is a core member of the MCC, which is timely assembled and disassembled through the conformational transition of 'O-MAD2' to 'C-MAD2' via a 'MAD2 template' model throughout the cell cycle (*De Antoni et al., 2005*). The disassembly of the MCC machinery is

**Table 2.** Summary of the variants identified after filtering by whole-exome sequencing (WES) in patients from three families (F1: II-1, F2: II-1, and F3: II-1).

|  | F1: II-1 | F2: II-1 | F3: II-1 |
| --- | --- | --- | --- |
| Total variants | 134,159 | 392,392 | 66,914 |
| After excluding variants reported in dbSNP, 1000 genomes, EVS, ExAC, and gnomAD (**MAF <0.01**) | 3619 | 14,023 | 22,832 |
| Exonic nonsynonymous or splicing variants, or coding indels | 273 | 571 | 9660 |
| Homozygous or compound heterozygous (excluding X and Y chromosomes) | 12 | 14 | 11 |
| Homozygous | 6 | 5 | 3 |
| In homozygous region >2 Mb | 1 | – | – |

The online version of this article includes the following source data for table 2:

**Source data 1.** Homozygous and compound heterozygous variants identified by whole-exome sequencing (WES) that survived filtering in patient F1: II-1, F2: II-1, and F3: II-1.

**Table 3.** *MAD2L1BP* pathogenic variants observed in the three families.

| Families | Genomic position | cDNA change | Protein change | Mutation type | SIFT* | PPH2* | Mutation taster* | NNsplice* | ASSP* | gnomAD[†] allele | gnomAD[†] homozygotes |
|---|---|---|---|---|---|---|---|---|---|---|---|
| 1 | Chr6: 43608202 | c.853C>T | p.R285* | Nonsense | NA | NA | D | NA | NA | 3/248598 | 0/248598 |
| 2 | Chr6: 43607867 | c.518delT | p.F173Sfs4* | Frameshift deletion | NA | NA | D | NA | NA | Not found | Not found |
| | Chr6: 43600837 | c.21-94G>A | – | Splicing | NA | NA | NA | 0.13>0.0[‡] | 7.1>2.2[‡] | 111/31250 | 2/31250 |
| 3 | Chr6: 43607890 | c.541C>T | p.R181* | Nonsense | NA | NA | D | NA | NA | 2/250850 | 0/250850 |

Abbreviations are as follows: D, deleterious; NA, not available.

*Mutation assessment by SIFT, Polyphen-2 (PPH2), Mutation Taster, NNsplice, and ASSP.

[†]Frequency of corresponding mutations in all population of gnomAD.

[‡]The variant c.21-94G>A is predicted to introduce an alternative splice acceptor by NNsplice (the increase of the acceptor site score from 0 up to 0.13) and by ASSP (the increase of the acceptor site score from <2.2 up to 7.1).

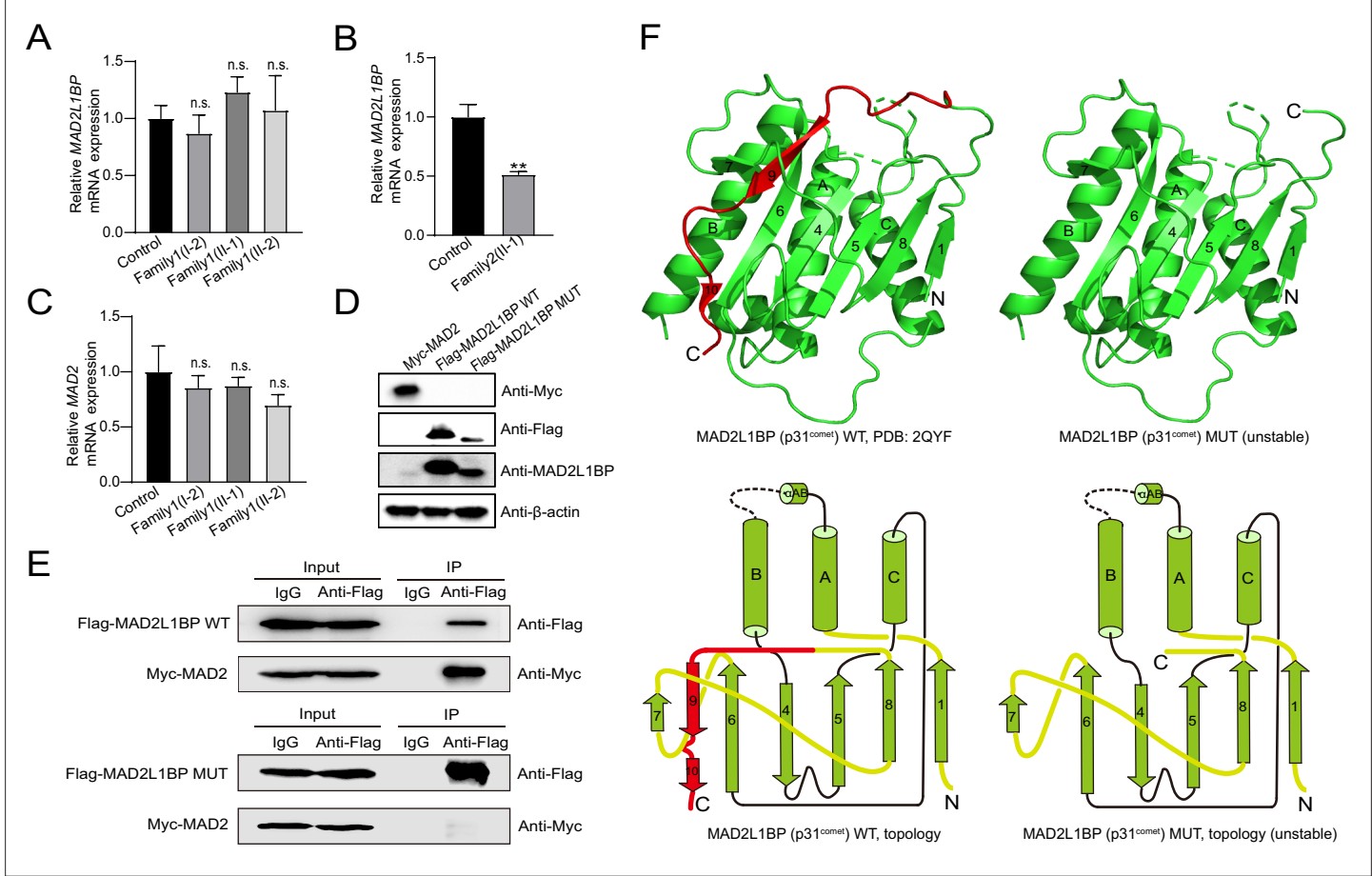

**Figure 2.** Adverse impacts of *MAD2L1BP* variants on mRNA expression and protein function. (**A–B**) Quantitative PCR (qPCR) assays comparing the mRNA expression levels of *MAD2L1BP* in peripheral blood samples from Family 1 and Family 2 as indicated. The experiments were performed in technical triplicates. (**C**) qPCR assay showing the mRNA expression levels of *MAD2* in individual blood samples from Family 1. The experiments were performed in technical triplicates. (**D**) The ectopic protein expression levels of Myc-MAD2, Flag-MAD2L1BP WT(wild-type), and Flag-MAD2L1BP MUT (p.R285*) by immunoblotting following transfection of individual plasmids into 293T cells in culture. Full-length and truncated MAD2L1BP protein can be seen in the panel. (**E**) Co-immunoprecipitation (Co-IP) assays demonstrating that the truncated MAD2L1BP MUT (p.R285*) lost interaction with MAD2, as compared with MAD2L1BP WT, when co-transfected into 293T cells in vitro. (**F**) Ribbon and topological diagrams showing the structure of MAD2L1BP WT (PDB ID: 2QYF) and the predicted MAD2L1BP MUT (p.R285*). Notably, MAD2L1BP MUT (p.R285*) lacks a C-terminal seatbelt configuration (highlighted in red) resulting in structural instability.

The online version of this article includes the following source data and figure supplement(s) for figure 2:

**Source data 1.** Blots for detecting Myc-MAD2, Flag-MAD2L1BP WT, Flag-MAD2L1BP MUT protein expression.

**Source data 2.** Co-immunoprecipitation (Co-IP) blots for detecting interaction between MAD2 and MAD2L1BP WT or MAD2L1BP MUT.

**Source data 3.** The original file of blots for *Figure 2D*.

**Source data 4.** The original file of blots for *Figure 2E*.

**Figure supplement 1.** Phylogenetic tree of *MAD2L1BP* and mRNA expression pattern of *MAD2L1BP* and *MAD2* in human follicles and mouse oocytes.

**Figure supplement 2.** Minigene assay showing the aberrant alternative splicing of *MAD2L1BP* exons associated with the variant c.21-94G>A from the Family 2.

predominantly achieved through MAD2L1BP, which acts as an adaptor that binds MAD2 and recruits ATPase TRIP13 to the MCC (*Alfieri et al., 2018*; *Ma and Poon, 2016*; *Wang et al., 2014*). Compelling studies have shown that MAD2 overexpression or deficient TRIP13 rendered oocyte meiotic arrest in both mice and humans, suggesting an indispensable role of MAD2-MAD2L1BP-TRIP13 signaling in driving meiotic division (*Wassmann et al., 2003*). We therefore subsequently carried out co-immunoprecipitation (Co-IP) assays using Myc-tagged MAD2 plasmid, and Flag-tagged wild-type (WT) and p.R285* mutated MAD2L1BP constructs (*Figures 1B and 2D*). Not surprisingly, while the Flag-tagged

WT MAD2L1BP could strongly pull down the full-length MAD2 as judged by Co-IP, the loss of 22 aa in MAD2L1BP at the C-terminus completely abolished its binding to MAD2, suggesting an essential function of the C-terminal 22 aa in stabilizing the binding of MAD2L1BP to MAD2 (*Figure 2E*). In support of this finding, the resolved crystal structure of MAD2L1BP (PDB: 2QYF) showed that its C-terminal 22 residues fold into the β9 and β10 strands, which are in charge of the physical interaction between MAD2L1BP and MAD2 like a seatbelt essential for the conformational stability (*Yang et al., 2007*; *Figures 1B and 2F*). This evidence altogether suggests that the C-terminus of MAD2L1BP is required for its interaction with MAD2, and the oocyte maturation arrest in the female patients was attributed to the truncated MAD2L1BP variants identified.

## Overexpression of wild-type Mad2l1bp, but not mutant Mad2l1bp, rescued the mouse oocyte MI arrest in vitro

To further corroborate the causal relationship between MAD2L1BP variants and female infertility resulting from oocyte MI arrest, we next interrogated the phenotypic outcome by overexpression of wild-type (WT) or mutant *Mad2l1bp* (hereafter all 'Mut' represents 'p.R285*') via cRNA injection into PMSG-primed mouse oocytes at the GV stage (*Figure 3A*). As shown in *Figure 3B and C*, exogenous expression of either WT or p.R285* *Mad2l1bp* alone resulted in similar percentages of oocytes undergoing GVBD. However, further examination in detail showed that the oocytes supplemented with WT *Mad2l1bp* extruded the polar body much earlier than those supplemented with p.R285* *Mad2l1bp* injection. A quantitative comparison from the time-lapse imaging experiments revealed that microinjection of the WT *Mad2l1bp* cRNAs accelerated the polar body extrusion (PBE) by ~3 hr as compared with those in the control groups (non-injected or GFP cRNA group), although the resultant percentages of oocytes with PBE were comparable among WT *Mad2l1bp* and control groups (~80%) (*Figure 3B and D*). By comparison, supplementation with p.R285* *Mad2l1bp* cRNAs not only postponed the extrusion of the polar body but also significantly lowered the final PBE rate in the oocytes (~40%) (*Figure 3B and D*). This evidence reinforces that the mutant p.R285* is a deleterious variant underlying oocyte MI arrest, which somehow implicates dominant-negative effect on the full-length WT Mad2l1bp. Given that the truncated Mad2l1bp protein is unlikely to be folded resulting in its inability to interact with its known binding partners in somatic cells, we reason that there likely exist potential factors that interplay with the mutant Mad2l1bp protein in the oocytes.

Subsequently, we eliminated endogenous Mad2l1bp followed by the 'rescue' experiment through microinjection of exogenous *Mad2l1bp* cRNAs. To this end, we first designed three specific 'siRNAs' against *Mad2l1bp* in order to knock down the *Mad2l1bp* mRNA levels in oocytes. Although we have repeatedly validated that one of the three siRNAs was able to reduce the *Mad2l1bp* mRNA levels by 70% in mouse F9 cells in vitro, this siRNA failed to deplete *Mad2l1bp* mRNA significantly after injection into the oocytes (data not shown). Next, we adopted an alternative strategy by overexpression of *Mad2* mRNA inducing meiotic MI arrest in GV oocytes and monitored the meiotic progression during later in vitro culture in M16 medium (*Wassmann et al., 2003*). We thus transcribed the *Mad2* cRNAs in vitro and injected the *Mad2* cRNAs into oocytes at GV stage. As expected, almost all the oocytes from the GV stage were trapped at the MI stage with no polar body extrusion under extended culture in M16 medium (*Figure 3E*, upper panel). This is consistent with previous studies showing that Mad2 overexpression stimulated the meiotic spindle checkpoint response, culminating in MI arrest in the oocytes. Using this model, we next simultaneously co-injected the *Mad2* cRNAs combined with *Mad2l1bp* cRNAs (WT vs p.R285* mutant) generated after transcription from the linearized plasmids. Consistently, the majority of GV oocytes supplemented with the WT *Mad2l1bp* cRNAs developed beyond meiosis I, as judged by the PBE (*Figure 3E and F*). In contrast, the oocytes supplemented with the p.R285* *Mad2l1bp* cRNAs rarely extruded the polar body, suggesting that the oocyte maturation was impeded at the MI stage, and the MAD2L1BP variant lost its function in vivo as compared with its WT counterpart (*Figure 3E and F*).

Moreover, we also examined the SAC integrity and meiotic progression by immunofluorescence co-staining with Bub3 (MCC component) and CREST (kinetochore) following the published procedures (*Karasu et al., 2019*; *Aboelenain et al., 2022*; *Mihajlović et al., 2021*; *Touati et al., 2015*; *Figure 3— figure supplement 1*). In the oocytes co-injected with p.R285* *Mad2l1bp cRNA*, we observed persistent Bub3-positive staining in the kinetochore of bivalent chromosome spreads throughout in vitro maturation, indicating activated SAC response in the presence of p.R285* *Mad2l1bp variant*

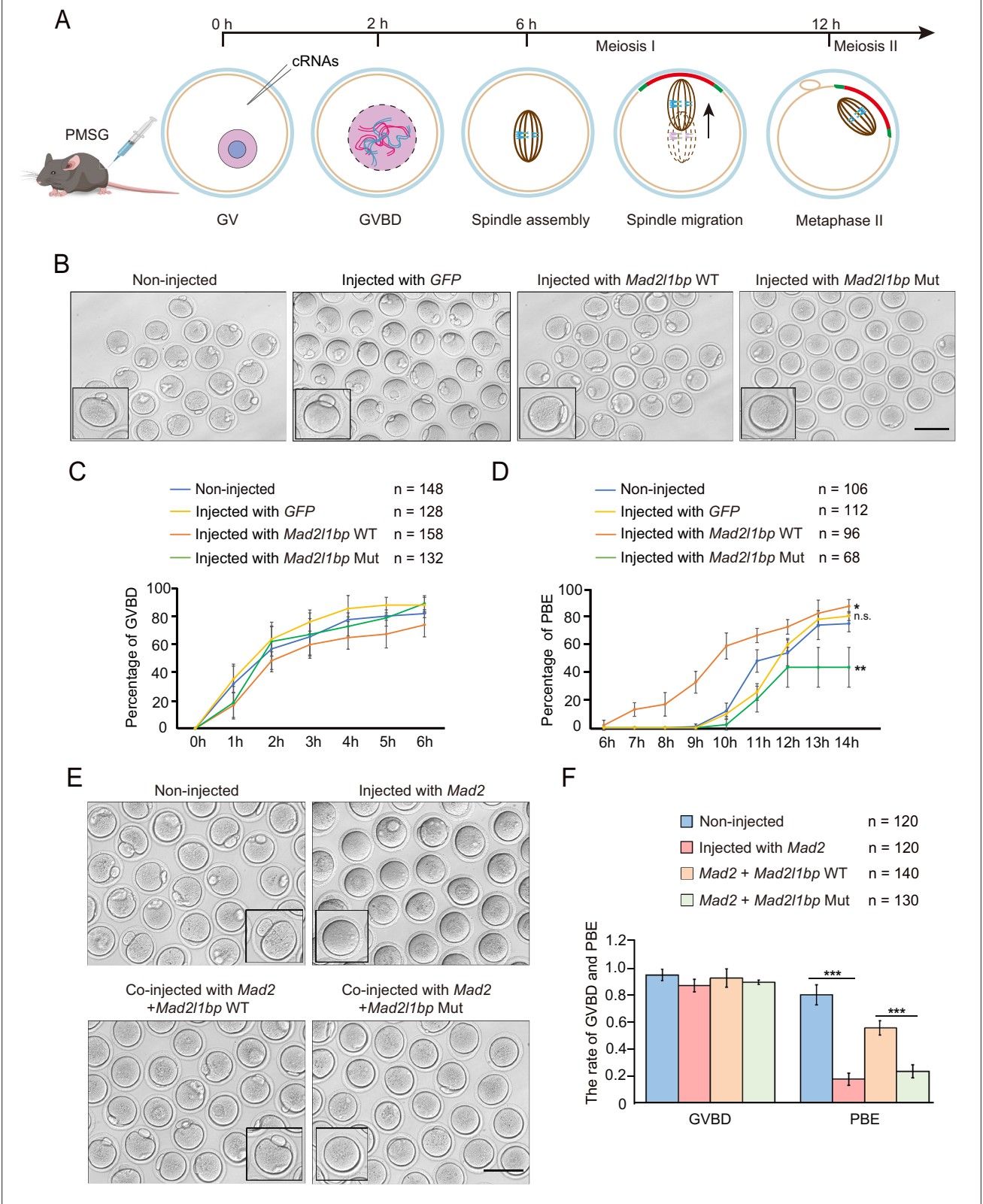

**Figure 3.** cRNA microinjection of *Mad2l1bp* accelerated or rescued the meiotic division in mouse germinal vesicle (GV) oocytes. (**A**) A schematic diagram depicting the time points of distinctive events through meiotic progression in mouse oocytes. cRNAs were microinjected into GV oocytes after 44 hr of PMSG priming. (**B**) Representative morphology of oocytes after microinjection with cRNAs encoding mock *GFP*, mouse *Mad2l1bp WT*, and *Mad2l1bp Mut* (equivalent to human p.R285*). Non-injected group was treated similarly to the other three groups without cRNA microinjection.

*Figure 3 continued on next page*

*Figure 3 continued*

Images were taken 14 hr following release from milrinone inhibitor. Scale bar, 100 μm. (**C and D**) Kinetic recordings showing the percentages of oocytes with GV breakdown (GVBD) (**C**), and the oocyte polar body extrusion (PBE) rate (**D**) through the time-lapse imaging experiment. The total numbers of oocytes microinjected were labeled as indicated. The experiments were performed in biological triplicates. Data are presented as mean ± SEM. (**E**) Representative morphology of oocytes after cRNA co-microinjection of Mad2 supplemented with WT or *Mad2l1bp* Mut (equivalent to human p.R285*) into the mouse GV oocytes. Images were taken 14 hr after release from milrinone. Scale bar, 100 μm. (**F**) Comparison of the percentages of oocytes with GVBD and PBE among the four microinjection groups in (**E**). The total numbers of oocytes microinjected were labeled as indicated. The experiments were performed in biological triplicates. The statistics were performed with *Student's t-test*. Data are presented as mean ± SEM. '***' indicates p<0.001.

The online version of this article includes the following source data and figure supplement(s) for figure 3:

**Source data 1.** cRNA microinjection of full-length or truncated *Mad2l1bp* uncovered their discordant roles in driving the extrusion of polar body 1 (PB1) in mouse oocytes.

**Source data 2.** Co-injection of *Mad2* cRNAs with *Mad2l1bp* cRNAs (WT vs p.R285* mutant) indicated the Mad2l1bp variant lost its function in vivo as compared with its WT counterpart.

**Figure supplement 1.** Mad2l1bp variant lost its function in silencing the spindle assembly checkpoint (SAC) in oocytes.

(***Figure 3—figure supplement 1***; ***Li et al., 2009***). In contrast, the oocytes co-injected with *Mad2l1bp* WT predominantly harbored univalent chromosomes (~85%) and much lower intensity of Bub3-positive staining, indicative of the SAC silence and completion of meiotic division I (***Figure 3—figure supplement 1***). Together, these results reinforced the detrimental effect of *MAD2L1BP variant* on SAC silence accounting for female infertility.

## Rescue of human oocyte meiotic arrest by microinjection of *MAD2L1BP* cRNA

The microinjection experiments performed in mouse oocytes as described above corroborated the *loss-of-function* effect of p.R285* *MAD2L1BP* variant. We next sought to determine whether we can recapitulate similar results in the frozen oocytes from the patients. Six oocytes from the first patient (F1: II-1) were exploited for the microinjection of full-length *MAD2L1BP* cRNAs. Intriguingly, we attained four oocytes with PB1 after culture in G-MOPS medium for in vitro maturation, indicative of the successful completion of meiosis I in the patient's oocytes when supplemented with exogenous intact *MAD2L1BP* cRNAs (***Figure 4A***). In contrast, the other two recovered oocytes with sham microinjection failed to extrude the polar body as a control (***Figure 4A***). This assay provided evidence supporting the p.R285* *MAD2L1BP* variant as a causative variant underlying oocyte MI arrest in the patient, and hints a possible treatment strategy by supplementation with exogenous *MAD2L1BP* cRNAs. We were not able to conduct the similar assay using the oocytes from the other two patients due to the limited number of mutant oocytes recovered. Of note, the brother of the female patient from Family 1 (***Figure 1A***) was also infertile. Examination of his semen sample showed that he had a markedly declined number of sperm, some of which exhibit aberrant head morphology, reminiscent of a possible role of MAD2L1BP in spermatogenesis as well (***Figure 1—figure supplement 2***, ***Supplementary file 1A***).

In vitro functional assays in mouse oocytes as well as rescue studies in human patient's oocytes, as shown above, demonstrate the essential role of MAD2L1BP in driving oocyte meiotic division. Next, we asked how the molecular transcriptome was impacted in the patient's oocytes carrying *p.R285** MAD2L1BP. To this end, we carried out an optimized single-cell SMART-seq2 in duplicates using two frozen MI oocytes from healthy female donors and two remaining frozen MI oocytes from the first patient (F1: II-1). Bioinformatic analysis revealed that the numbers of expressed transcripts in healthy control and MAD2L1BP[R285*] oocytes were comparable, with an average of approximately $1.4 \times 10^4$ detected (***Figure 4—figure supplement 1A***). Among them, we discovered that 1811 transcripts were down-regulated while 1705 transcripts were up-regulated in MAD2L1BP[R285*] oocytes (***Figure 4—figure supplement 1B***). The Gene Ontology (GO) analysis revealed that, in support of the phenotypic meiotic arrest, the down-regulated transcripts were mainly engaged in pathways related to cell division and cell cycle (***Figure 4—figure supplement 1C***). Interestingly, the transcript expression levels for MCC components, including MAD2, BUBR1, and CDC20 (except BUB3), as well as TRIP13 were comparable between MAD2L1BP[R285*] and control oocytes (***Figure 4—figure supplement 1E***), which is consistent with previous studies and our qPCR experiments (***Figure 2C***; ***Ma and Poon, 2016***).

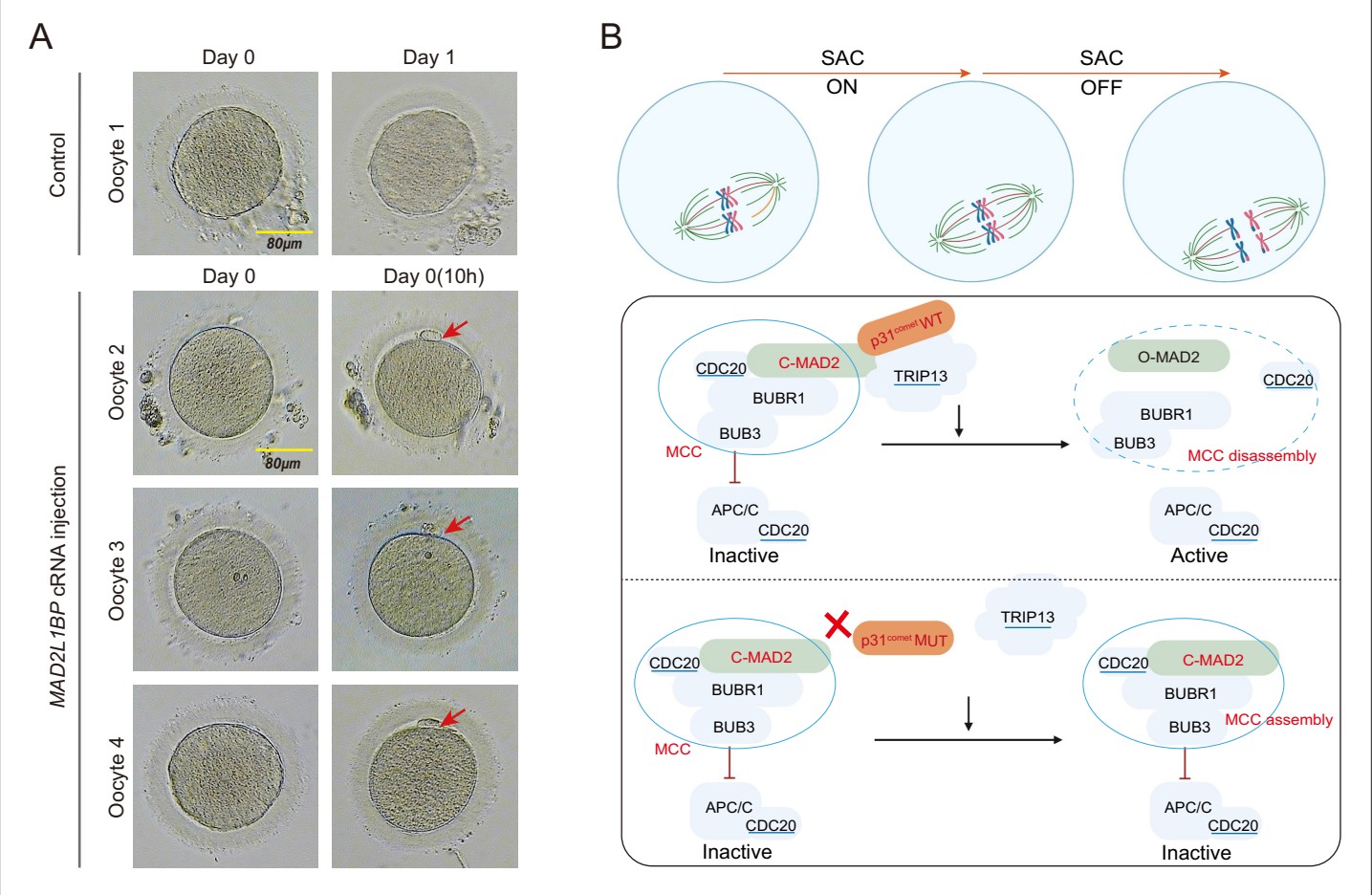

**Figure 4.** Meiotic rescue of the metaphase I (MI)-arrested frozen human oocytes by *MAD2L1BP* cRNA microinjection. (**A**) The resumption of polar body extrusion through *MAD2L1BP* cRNA microinjection into the frozen oocytes of the patient in Family 1 (II-1). The extrusion of the first polar body indicates the completion of meiosis I of the oocytes. In the control group, the retrieved MI oocytes remained arrested at MI after sham microinjection (upper panel). Six oocytes from the patient in Family 1 (II-1) were microinjected with full-length *MAD2L1BP* cRNAs, of which four successfully finished extrusion of polar body 1 (PB1) (lower panel). The red arrows point to PB1. Scale bar, 80 µm. (**B**) A summarized working model depicting how MAD2L1BP(p31comet) deficiency causes human oocyte meiotic arrest. The mitotic checkpoint complex (MCC) is well known to comprise four core members: BUBR1, BUB3, CDC20, and MAD2. MAD2L1BP(p31comet) is known as the core adaptor that bridges the interaction between MAD2 and TRIP13 (*Yang et al., 2007*; *Alfieri et al., 2018*). The MAD2L1BP (p31comet)-mediated interplay between MAD2 and TRIP13, known as the MAD2•MAD2L1BP(p31comet)•TRIP13 axis, disassembles the MCC signaling, which consequently drives the meiotic progression during oocyte meiosis. Top panel: The spindle assembly checkpoint (SAC) is switched 'ON' until all the chromosome kinetochores are correctly connected to the spindles. Bottom panel: In the presence of WT MAD2L1BP, SAC is turned 'OFF' through the recruitment of MAD2-MAD2L1BP-TRIP13 signaling to silence MCC, leading to the meiotic cell-cycle progression. By contrast, in the presence of mutated MAD2L1BP, MCC is unable to be silenced, resulting in persistent SAC activation and oocyte meiotic arrest.

The online version of this article includes the following figure supplement(s) for figure 4:

**Figure supplement 1.** Modified Smart-seq2 analysis revealed that MAD2L1BP variant impaired human oocyte meiotic progression.

However, the expression levels of other cell division-related factors were significantly decreased in MAD2L1BPR285* oocytes (*Figure 4—figure supplement 1F*). Intriguingly, the GO analysis unveiled that the up-regulated transcripts were predominantly relevant to mitochondrial metabolism, which is critical for energy homeostasis during oocyte maturation (*Richani et al., 2021*; *Trebichalská et al., 2021*; *Figure 4—figure supplement 1D and G*). We reasoned that the aberrantly accumulated mitochondrial transcripts might stem from the secondary defects owing to the failure of timely meiotic cell-cycle transition in MAD2L1BPR285* oocytes. Altogether, these results using human patient's oocytes provided evidence that *MAD2L1BP* is a bona fide essential gene in driving human oocyte meiotic division I.

## Discussion

To ensure the fidelity of chromosome segregation, the cells, including the oocytes, have evolved an exquisite surveillance machinery – the SAC, which represses the downstream anaphase-promoting complex (APC/C) activity until all chromosomes are correctly aligned and attached to the bipolar spindles through the kinetochores (*Mihajlović and FitzHarris, 2018*; *Liu and Zhang, 2016*; *Charalambous et al., 2023*; *Vallot et al., 2018*; *Figure 4B*). Central to the SAC is the physical presence of the MCC, which is comprised of four core components: BUBR1, MAD2, BUB3, and the APC activator CDC20. These players are phylogenetically conserved in metazoan species, and mouse models have provided unambiguous evidence showing their indispensable roles in cell-cycle progression (*Liu and Zhang, 2016*; *Lara-Gonzalez et al., 2021*). In response to the kinetochores improperly attached to the spindles in the prometaphase, the SAC will be activated, which elicits a hierarchical recruitment of proteins to the outer kinetochores (*Mihajlović and FitzHarris, 2018*; *Charalambous et al., 2023*; *Vallot et al., 2018*; *Figure 4B*). This drives the conversion of Mad2 from its inactive, open conformer (O-Mad2) to its closed, active form (C-Mad2) through a generally accepted 'Mad1-Mad2 templating' mechanism for cytosolic propagation of spindle checkpoint signal (*Lara-Gonzalez et al., 2021*). Genetic mouse models have shown that Mad2 is the core player of SAC signaling – inactivation of Mad2 caused the accelerated mitotic exit whereas overexpression of Mad2 induced precocious activation of SAC in somatic cells and meiotic MI arrest in oocytes, suggesting MAD2 is functional and required for normal oocyte meiotic progression (*Wassmann et al., 2003*; *Lara-Gonzalez et al., 2021*; *Niault et al., 2007*). Once the kinetochore-spindle attachment is satisfactorily achieved, the inhibitory signaling from the SAC must be timely quenched to allow anaphase progression. This is primarily achieved through MAD2 recognition by MAD2L1BP, which further recruits AAA+ATPase, TRIP13, that binds and subsequently dissembles the MCC in an ATP-dependent manner (*Alfieri et al., 2018*; *Brulotte et al., 2017*). Conversely, deficient SAC disassembly, for example, *MAD2L1BP* mutations in this study, disrupts the MAD2•MAD2L1BP(p-31[comet])•TRIP13 axis, leading to sustained SAC activation and consequently the oocyte maturation arrest (*Figure 4B*, lower panel).

Intriguingly, recent studies have identified pathogenic biallelic variants in *TRIP13* causing female primary infertility owing to the oocyte maturation arrest predominantly at MI stage (*Zhang et al., 2020*). Furthermore, a spectrum of deleterious variants in *CDC20*, the core component of MCC, have also been discovered, that account for human oocyte meiotic arrest and early embryonic failure (*Zhao et al., 2020*). To the best of our knowledge, this study is the first report that identified and characterized causative *loss-of-function* variants in *MAD2L1BP* underlying human oocyte maturation arrest. Consistent with the abnormalities in *MAD2L1BP*-null oocyte meiotic division, *MAD2L1BP* KO cells exhibited mitotic delay and lagging chromosomes (*Ma and Poon, 2016*), but some of them managed to reinitiate cell cycle, which is distinct from the complete arrest in the mutant oocytes. Notably, the younger brother of the female patient from Family 1 (*Figure 1A*) was also infertile. The significantly declined number and the aberrant morphology of the sperm retrieved appear to implicate that there might be a meiotic disruption in this male patient. Unfortunately, after multiple attempts, we were not able to obtain a testicular biopsy for further pathological analysis. Hence, further exploration of the roles of MAD2L1BP variants in the male meiosis would be an interesting direction in the future.

Moreover, in order to recapitulate the deleterious effect of the human *MAD2L1BP* variants in mice, we have attempted to generate the knock-in (KI) mouse model that mimic the equivalent human mutations. However, *Mad2l1bp*-deficient mice died perinatally owing to the neonatal hypoglycemia resulting from the liver glycogen shortage, rendering us fail to get the homozygous KI offspring (*Choi et al., 2016*). Taken together, with a small-scale cohort of genetic screening in 50 oocyte MI-arrested patients, our study adds novel pathogenic variants of *MAD2L1BP* to the genetic mutational spectrum underlying human oocyte maturation arrest, and likely provides novel diagnostic and therapeutic avenues for primary female infertility in the future.

# Materials and methods

**Key resources table**

| Reagent type (species) or resource | Designation | Source or reference | Identifiers | Additional information |
|---|---|---|---|---|
| Strain, strain background (*Escherichia coli*) | DH5α competent *E. coli* | Thermo Fisher | Cat # 18265017 | |
| Genetic reagent (*Mus musculus*) | C57BL/6J | Jackson Laboratories | RRID: IMSR_JAX:000664 | |
| Cell line (*Homo sapiens*) | HEK293T cells | ATCC | Cat # CRL-3216; RRID: CVCL_0063 | Medium: DMEM+10% FBS |
| Recombinant DNA reagent | pcDNA3.1-human MAD2L1BP isoform1 | This paper | pcDNA3.1-hMAD2L1BP iso1 | Synthesized from Genewiz. Available from Jianqiang Bao's lab |
| Recombinant DNA reagent | pcDNA3.1-human MAD2L1BP isoform2 | This paper | pcDNA3.1-hMAD2L1BP iso2 | Synthesized from Genewiz. Available from Jianqiang Bao's lab |
| Recombinant DNA reagent | M46-mouse Mad2l1bp full-length-Flag | This paper | M46-mMad2l1bp-Flag | Available from Jianqiang Bao's lab |
| Recombinant DNA reagent | p3xFLAG-CMV-24-mouse Mad2l1bp truncated (1-254aa, equivalent to human p. R285*) | This paper | p3xFLAG-mMad2l1bp Mut | Available from Jianqiang Bao's lab |
| Recombinant DNA reagent | p3xFLAG-CMV-24-mouse Mad2l1bp full-length(1-276aa) | This paper | p3xFLAG-mMad2l1bp WT | Available from Jianqiang Bao's lab |
| Recombinant DNA reagent | pcDNA3.1-mouse Mad2 full-length-Myc | This paper | pcDNA3.1-mMad2-Myc | Available from Jianqiang Bao's lab |
| Recombinant DNA reagent | pcDNA-human MAD2L1BP-WT minigene | This paper | pcDNA-hMAD2L1BP-WT minigene | Available from Jianqiang Bao's lab |
| Recombinant DNA reagent | pcDNA-human MAD2L1BP-c.21-94G>A minigene | This paper | pcDNA-hMAD2L1BP-c.21–94G>A minigene | Available from Jianqiang Bao's lab |
| Antibody | Anti-MAD2L1BP (Mouse monoclonal) | Santa Cruz | Cat # sc-134381 | WB (1:200) |
| Antibody | Anti-Flag (Mouse monoclonal) | Proteintech | Cat # 66008-3-Ig | WB (1:2000) |
| Antibody | Anti-Myc (Mouse monoclonal) | Proteintech | Cat # 60003-2-Ig | WB (1:5000) |
| Antibody | Anti-Beta Actin (Mouse monoclonal) | Proteintech | Cat # 66009-1-Ig | WB (1:10,000) |
| Antibody | HRP-conjugated goat anti-mouse IgG heavy chain (Goat polyclonal) | ABclonal | Cat # AS064 | WB (1:10,000) |
| Antibody | HRP-conjugated goat anti-mouse IgG light chain (Goat polyclonal) | ABclonal | Cat # AS062 | WB (1:10,000) |
| Antibody | Anti-BUB3 (Rabbit polyclonal) | Santa Cruz | Cat # sc-28258 | IF (1:300) |
| Antibody | Anti-CREST (Human, unknown clonality) | Fitzgerald Industries International | Cat # 90C-CS1058 | IF (1:100) |
| Antibody | Goat Anti-Human IgG H&L (FITC) (Goat polyclonal) | ZENBIO | Cat # 550022 | IF (1:500) |
| Antibody | CoraLite 594 Anti-Rabbit (Goat polyclonal) | Proteintech | Cat # SA00013-4 | IF (1:500) |
| Software, algorithm | R software | R Project for Statistical Computing | RRID:SCR_001905 http://www.r-project.org/ | |

*Continued on next page*

*Continued*

| Reagent type (species) or resource | Designation | Source or reference | Identifiers | Additional information |
|---|---|---|---|---|
| Software, algorithm | DAVID | DAVID | RRID:SCR_001881 https://david.ncifcrf.gov/ | |
| Software, algorithm | Fiji/ImageJ | Fiji | RRID:SCR_002285 http://fiji.sc | |
| Software, algorithm | Jalview | Jalview | RRID:SCR_006459 https://www.jalview.org/ | |
| Software, algorithm | Prism | GraphPad Prism | RRID:SCR_002798 http://www.graphpad.com/ | |
| Software, algorithm | PyMOL | PyMOL | RRID:SCR_000305 http://www.pymol.org/ | |
| Other | GenBank | NIH | RRID:SCR_002760 https://www.ncbi.nlm.nih.gov/genbank/ | Database |
| Other | gnomAD | Genome Aggregation Database | RRID:SCR_014964 http://gnomad.broadinstitute.org/ | Database |
| Other | OMIM | OMIM | RRID:SCR_006437 http://omim.org | Database |
| Other | ASSP | ASSP | http://wangcomputing.com/assp/ | Web resource |
| Other | EvolView | EvolView | http://evolgenius.info | Web resource |
| Other | Mutation Taster | MutationTaster | RRID:SCR_010777 http://www.mutationtaster.org/ | Web resource |
| Other | NNSplice | NNSplice | http://www.fruitfly.org/seq_tools/splice.html | Web resource |
| Other | PolyPhen-2 | PolyPhen: Polymorphism Phenotyping | RRID:SCR_013189 http://genetics.bwh.harvard.edu/pph2/ | Web resource |
| Other | SIFT | SIFT | RRID:SCR_012813 https://sift.bii.a-star.edu.sg/ | Web resource |

## Patients

A total of 50 primary infertile females undergoing recurrent IVF/ICSI failure due to complete oocyte maturation arrest were recruited between July 2014 and October 2021, from the First Affiliated Hospital of USTC (University of Science and Technology of China) and the Hospital for Reproductive Medicine, Shandong University. All of them had a normal karyotype (46, XX). Peripheral blood samples from all affected individuals and their available family members and 10 MI arrested oocytes from the patient (F1: II-1) were donated for this study with written informed consent. This study was approved by the biomedical research ethics committees of Anhui Medical University on March 1, 2017 (reference number 20170121; the Anhui Provincial Hospital Affiliated to Anhui Medical University, now renamed as the First Affiliated Hospital of USTC after December 2017).

## WES and data analysis

WES was performed using the DNA extracted from the periphery blood samples of the probands and the family members. Exonic DNA libraries were prepared using the Agilent Human SureSelect All Exon V6 kit and exome sequencing was performed on an Illumina NovaSeq 6000 platform. Clean sequencing reads were aligned to the human reference sequence (hg19). Sequence variants, including single-nucleotide variants (SNVs) and small insertions or deletions (indels), were annotated by the ANNOVAR pipeline. Common variants (defined as a MAF above 1% in public databases: 1000 genome, dbSNP, ESP6500, gnomAD, or ExAC) were excluded. SNVs and indels were classified by

position as intergenic, 5' UTR, 3'UTR, intronic, splicing, or exonic. Exonic variants were then classified by predicted amino acid change as a stopgain, missense, synonymous, frameshift, indel or inframe, or possible splicing variants. For coding or possible splicing variants, the conservation at the variant site and the potential effect on protein function were evaluated with *in silico* tools: SIFT, PolyPhen-2, MutationTaster, NNSplice, and ASSP (*Table 3*). Initially, we filtered genes that had predicted deleterious variants in at least two unrelated affected women and had not been previously reported (*Table 2—source data 1*).

## Sanger sequencing of the candidate variants

The DNA samples were extracted from the peripheral blood from the patients and their available family members. Sanger sequencing of all the coding regions, 200 bp of the flanking intron splicing sites and the promoter region (1223 bp upstream of the start codon) of *MAD2L1BP,* were performed in the three probands and all their available family members. qPCR analysis of all the coding regions of *MAD2L1BP* was also performed to screen for small copy number variants in the patient (F2:II-1) and her family members. See primers in *Supplementary file 1B*.

## Plasmid construction

Total RNA was extracted from 10 ovaries from female mice between 4 and 6 weeks of age using TRIzol following the manufacturer's protocol. First-strand cDNA was synthesized using a ProtoScript II cDNA first strand kit (NEB) with 1 µg of total RNA. The full-length and truncated coding sequences of mouse *Mad2l1bp* were amplified and cloned into the p3xFLAG-CMV-24 vector. cDNA fragment encoding mouse *Mad2* was PCR-amplified and inserted into the pcDNA3.1 vector with a Myc tag. Human *MAD2L1BP* coding sequence were synthesized from Genewiz and cloned into the pcDNA3.1 vector. All construct sequences were verified by Sanger sequencing.

## Real-time qPCR

Total RNA was extracted using a SPARKeasy Frozen whole blood total RNA Kit; 0.5 µg RNA from each sample was used for reverse transcription with a RevertAid First Strand cDNA Synthesis Kit (Thermo Scientific). qPCR was performed with ChamQ Universal SYBR qPCR Master Mix (Vazyme) using a Roche LightCycler 96 machine. Relative mRNA levels were calculated by normalizing to the levels of internal GAPDH control. The qPCR primers are shown in *Supplementary file 1B*.

## Western blotting

Protein extracts were denatured by heating for 10 min at 95°C in SDS-PAGE sample loading buffer. Proteins were separated by gel electrophoresis, followed by wet transfer to polyvinylidene difluoride membranes (Millipore). After blocking with 5% nonfat milk diluted in phosphate buffered saline supplemented with 0.05% Tween 20 for 1 hr, membranes were probed with primary antibodies against MAD2L1BP (1:200, Santa Cruz, Cat #sc-134381), Flag (1:2000, Proteintech, Cat #66008-3-Ig), and Myc (1:5000, Proteintech, Cat #60003-2-Ig). Beta-actin (1:10,000, Proteintech, Cat # 66009-1-Ig) was used as internal control. The secondary antibodies were HRP-conjugated goat anti-mouse IgG heavy chain (1:10,000, ABclonal, Cat #AS064) or HRP-conjugated goat anti-mouse IgG light chain (1:10,000, ABclonal, Cat #AS062). Target proteins were detected using the ECL Western Blotting Detection Kit (Tanon) according to the manufacturer's recommendation.

## Cell culture, plasmid transfection, and Co-IP

293T cells were maintained in high-glucose DMEM (Gibco) medium supplemented with 10% fetal bovine serum (FBS; Gemini) and 1% penicillin-streptomycin solution (Gibco) at 37°C in a humidified 5% $CO_2$ incubator. The identity of the cell line was authenticated regularly by short tandem repeat assay and routinely tested for the presence of *Mycoplasma* using qPCR assay. Transient transfections were performed using PEI reagent (Polyscience). Cells were collected 48 hr post transfection.

Immunoprecipitation assays were performed using 293T cells transfected with Flag-Mad2l1bp or Flag-Mad2l1bp Mut and Myc-Mad2 plasmids as described above. Total protein lysates from transfected cells were prepared in lysis buffer (20 mM Tris-HCl [pH 7.5], 200 mM NaCl, 1% NP-40, and 1% protease inhibitor cocktail [APE-BIO]), and precleared with protein A Dynabeads (Invitrogen) and protein G Dynabeads (Invitrogen) for 1 hr at 4°C. The precleared extracts (10% for input) were

incubated with Flag antibody or mouse IgG overnight at 4°C. Protein A and protein G were added to the antibody-extracts mixture followed by incubation for 1 hr at 4°C. The beads were washed with lysis buffer for four times prior to elution with SDS sample buffer. Western blotting was conducted as described above.

## In vitro transcription and preparation of cRNAs for microinjections

To prepare cRNAs for human oocyte microinjection, constructs were linearized with the Not1 restriction enzyme, and the linearized products were purified by phenol (Tris-saturated):chloroform extraction and ethanol precipitation, and then dissolved in nuclease-free water. The purified DNA templates were transcribed in vitro using the T7 High Yield RNA Transcription Kit (Novoprotein, E131). Transcribed cRNAs were capped using the Cap1 Capping System (Novoprotein, M082) and poly(A) tails (~100 bp) were added using the *E. coli* Poly(A) Polymerase (Novoprotein, M012). cRNAs were purified by phenol(water-saturated):chloroform extraction and ethanol precipitated, and resuspended in nuclease-free water.

To prepare cRNAs for mouse oocyte microinjection, expression vectors were linearized and subjected to phenol (Tris-saturated):chloroform extraction and ethanol precipitation. The linearized DNAs were in vitro transcribed using the T7 message mMACHINE Kit (Invitrogen, AM1344). Transcribed mRNAs were added with poly(A) tails (~200–250 bp) using the Poly(A) Tailing Kit (Invitrogen, AM1350), recovered by lithium chloride precipitation, and resuspended in nuclease-free water.

## Human oocyte collection and microinjection

Human oocytes (from control and affected individuals) were donated following informed consent. Oocytes from Family 1 (II-1) were recovered and randomly divided into the control and cRNA-injected groups. About 5 pl *MAD2L1BP* cRNA solution (1000 ng/µl) was microinjected into the oocytes. After injection, oocytes were cultured for in vitro maturation and considered to be matured when PB1 was extruded.

## Mouse oocyte culture and microinjection

Animal care and experimental procedures were conducted in accordance with the Animal Research Committee guidelines of Zhejiang University (approval # ZJU20210252 to HYF) and USTC (approval # 2019 N(A)-299 to JQB). The 21- to 23-day-old female mice were injected with 5 IU of PMSG and euthanized after 44 hr. Oocytes at the GV stage were harvested in M2 medium (Sigma-Aldrich, M7167) and cultured in mini-drops of M16 medium (Sigma-Aldrich, M7292) covered with mineral oil (Sigma-Aldrich, M5310) at 37°C in a 5% $CO_2$ atmosphere.

For microinjection, fully grown GV oocytes were incubated in M2 medium with 2 µM milrinone to inhibit spontaneous GVBD. All injections were performed using an Eppendorf transferman NK2 micromanipulator. Denuded oocytes were injected with 5–10 pl samples per oocyte. The concentration of all injected RNAs was adjusted to 1200 ng/µl. After injection, oocytes were washed and cultured in M16 medium at 37°C with 5% $CO_2$.

## Chromosome spreading and immunofluorescence

ZP-free oocytes were fixed in a solution containing 1% paraformaldehyde, 0.15% Triton X-100, and 3 mM DTT (Sigma-Aldrich) on glass slides for 30 min. After fixation, slides were air-dried. To perform immunofluorescence analysis of dispersed chromosomes, slides were blocked with 1% BSA and then incubated for 1 hr with primary antibodies. After rinsing three times, slides were incubated with secondary antibodies for 45 min, followed by DAPI staining. Slides were then mounted and examined with a Leica THUNDER Imager Live Cell with a K5 camera driven by the Leica Application Suite Software. Image processing was performed by ImageJ software. See antibodies information in Key resources table.

## Single-cell SMART-seq2 for RNA-seq library preparation

The single-oocyte SMART-seq2 protocol was based on the well-established SMART-seq2 protocol with in-house optimized modifications as indicated below. In brief, MI oocytes retrieved from healthy control or the first patient (F1: II-1) were washed at least five times in 1×PBS, and directly lysed in Lysis Buffer MasterMix (0.3% Triton X-100, 40 U/µl RNase inhibitor, 2.5 µM oligodT30VN, and 2.5 µM

dNTP mix). The oocyte lysis mixture was allowed to undergo at least one freeze-thaw cycle at –80°C to facilitate complete cytosolic lysis. The first strand cDNA synthesis was performed at 50°C for 60 min, followed by 16 cycles of PCR preamplification to achieve the full-length cDNA products through ISPCR primer-mediated semi-suppressive PCR. Fifty ng of size-selected full-length cDNAs was used for Tn5-guided library preparation using Hieff NGS Fast Tagment DNA Library Prep Kit for Illumina (Yeasen, 12207ES24). Final dual-barcoded libraries were achieved through PCR amplification for nine cycles with both index i5 and i7 primers prior to pooled library sequencing on the NovaSeq 6000 platform with PE150 mode (Novagene).

## RNA-seq data analysis

The sequencing raw reads were processed to remove adaptor contaminants and low-quality bases. The clean reads were aligned to the human genome (hg38) using STAR, and uniquely mapped reads were counted with RSEM by default parameters. We quantified gene expression levels with transcripts per million (TPM). For each sample, the expressed transcripts were defined with cutoff: TPM ≥1. Differentially expressed genes (DEGs) were assessed with the DESeq2 package with a cutoff: padj <0.05 and fold change (FC) ≥2. GO enrichment was performed using DAVID. Statistical analyses were performed using R software. The RNA-seq data in this work have been deposited in the NCBI Gene Expression Omnibus (GEO) under accession number GSE232488.

## Minigene assay

Minigene analysis was performed in 293T cells. Amplicons spanning exons 1–3 along with ~200 bp of the flanking intronic sequences were PCR-amplified using the genomic DNA from the normal control and were inserted into the pcDNA vector. Site-directed mutagenesis was performed to introduce the mutation c.21-94G>A. The minigene constructs (c.21-94G>A or WT) were transfected into cultured 293T cells using PEI reagent (Polyscience). Cells were harvested 48 hr after transfection and total RNA was extracted using TRIzol following the manufacturer's protocol. RT-PCR was performed using the indicated primers to amplify the target region (see *Supplementary file 1B* for primers sequences).

## Molecular modeling and evolutionary conservation analysis

MAD2L1BP mutations were mapped using PyMOL software according to the crystal structure of the Mad2/p31(comet)/Mad2-binding peptide ternary complex (PDB ID: 2QYF). Evolutionary conservation analysis was performed with Jalview software and processed on the Evolview website.

## Evaluation of oocyte phenotypes and sperm phenotypes

The stage of oocyte maturation was assessed under a light microscope as described previously (*Huang et al., 2018*). Semen analysis and papanicolaou staining were performed according to the protocol of the World Health Organization 2010 guidelines.

## Analysis of the expression pattern of MAD2L1BP and MAD2 in human follicles and mouse oocytes

RNA-seq data was downloaded from GSE71434 (*Zhang et al., 2016*) and GSE107746 (*Zhang et al., 2018*). Reads were mapped to the mouse genome (mm10) or human genome (hg38) using STAR. Relative expression quantification of genes and transcripts was performed using RSEM.

## Statistical analysis

Data are presented as the mean ± standard error of the mean (SEM). Most experiments were repeated at least three times. The results for two experimental groups were compared by two-tailed unpaired Student's t-tests. Statistically significant values of $p < 0.05$, $p < 0.01$, and $p < 0.001$ by two-tailed Student's t-test are indicated by asterisks (*), (**), and (***), respectively. 'n.s.' indicates nonsignificant.

## Materials availability statement

Plasmids constructed in this paper are available from Bao's laboratory upon request.

## Acknowledgements

This study was supported by the grants from National Natural Science Foundation of China (81801440, 82192874, 31970793, 32170856), the Ministry of Science and Technology of China (2019YFA0802600, 2022YFC2702600), Natural Science Foundation of Anhui Province (1808085MH241), the Fundamental Research Funds for the Central Universities (WK2070000156, WK9110000028, WK9100000032) and Startup funding (KY9100000001). The authors thank Prof. Lei Wang (Fudan University) for discussion with this manuscript. We thank Dr. Jianming Zeng (University of Macau), and all the members of his bioinformatics team, Biotrainee, for generously sharing their experience and codes, and the use of the biorstudio high-performance computing cluster (https://biorstudio.cloud) at Biotrainee and The Shanghai HS Biotech Co., Ltd for conducting the research reported in this paper.

## Additional information

### Funding

| Funder | Grant reference number | Author |
|---|---|---|
| National Natural Science Foundation of China | 81801440 | Lingli Huang |
| National Natural Science Foundation of China | 82192874 | Han Zhao |
| National Natural Science Foundation of China | 31970793 | Jianqiang Bao |
| National Natural Science Foundation of China | 32170856 | Jianqiang Bao |
| the Ministry of Science and Technology of China | 2019YFA0802600 | Jianqiang Bao |
| Fundamental Research Funds for the Central Universities | WK2070000156 | Jianqiang Bao |
| Startup funding | KY9100000001 | Jianqiang Bao |
| Natural Science Foundation of Anhui Province | 1808085MH241 | Lingli Huang |
| Fundamental Research Funds for the Central Universities | WK9110000028 | Lingli Huang |
| Fundamental Research Funds for the Central Universities | WK9100000032 | Jianqiang Bao |
| the Ministry of Science and Technology of China | 2022YFC2702600 | Jianqiang Bao |

The funders had no role in study design, data collection and interpretation, or the decision to submit the work for publication.

### Author contributions

Lingli Huang, Wenqing Li, Formal analysis, Validation, Investigation, Methodology, Writing – original draft; Xingxing Dai, Formal analysis, Validation, Methodology; Shuai Zhao, Formal analysis, Methodology; Bo Xu, Validation, Methodology; Fengsong Wang, Ren-Tao Jin, Xue Jiang, Yu Cheng, Jiaqi Zou, Methodology; Lihua Luo, Limin Wu, Investigation; Caoling Xu, Software; Xianhong Tong, Heng-Yu Fan, Resources; Han Zhao, Resources, Supervision; Jianqiang Bao, Conceptualization, Supervision, Writing – original draft, Writing - review and editing

### Author ORCIDs

Lingli Huang (iD) http://orcid.org/0000-0002-9748-267X

Heng-Yu Fan https://orcid.org/0000-0003-4544-4724
Jianqiang Bao https://orcid.org/0000-0003-1248-2687

### Ethics

Human subjects: Peripheral blood samples from all affected individuals and their available family members and ten Metaphase I (MI) arrested oocytes from the patient (F1: II-1) were donated for this study with written informed consent. This study was approved by the biomedical research ethics committees of Anhui Medical University on 1 March 2017(reference number 20170121; the Anhui Provincial Hospital Affiliated to Anhui Medical University, now renamed as the First Affiliated Hospital of USTC after December 2017).

Animal care and experimental procedures were conducted in accordance with the Animal Research Committee guidelines of Zhejiang University (approval # ZJU20210252 to H.Y.F) and USTC (approval # 2019-N(A)-299 to J.Q.B).

### Decision letter and Author response

Decision letter https://doi.org/10.7554/eLife.85649.sa1
Author response https://doi.org/10.7554/eLife.85649.sa2

## Additional files

### Supplementary files

- Supplementary file 1. The supplementary file includes file (A and B).
- MDAR checklist

### Data availability

Source Data files have been provided for Figure 2, Figure 3 and Table 2. Sequencing data have been deposited in GEO under accession code GSE232488.

The following dataset was generated:

| Author(s) | Year | Dataset title | Dataset URL | Database and Identifier |
|---|---|---|---|---|
| Li W, Xu C | 2023 | Biallelic variants in MAD2L1BP (p31comet) cause female infertility characterized by oocyte maturation arrest | http://www.ncbi.nlm.nih.gov/geo/query/acc.cgi?acc=GSE232488 | NCBI Gene Expression Omnibus, GSE232488 |

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
