## [Editor Report]

This important study identifies three independent patient mutations in MAD2L1BP (p31 comet) that cause infertility. Consistent with the known functions of p31 comet, convincing experiments in mouse oocytes imply that infertility could be caused by a failure to silence the spindle assembly checkpoint, though the mechanism was not determined. Although the sample size is small, a rescue experiment in human oocytes promises the potential for therapy.

---

## [Decision Letter]

**Decision letter after peer review:**

Thank you for submitting your article "Biallelic Variants in MAD2L1BP (p31 comet ) Cause Female Infertility Characterized by Oocyte Maturation Arrest" for consideration by *eLife*. Your article has been reviewed by 3 peer reviewers, and the evaluation has been overseen by a Reviewing Editor and Marianne Bronner as the Senior Editor. The following individual involved in the review of your submission has agreed to reveal their identity: Hongtao Yu (Reviewer #3).

Essential revisions:

1) In Figure 3D, injection of the Mad2l1bp mutant cRNA into mouse oocytes delayed polar body extrusion as compared to the un-injected control. This suggested that the Mad2l1bp mutant had a dominant-negative function. On the other hand, this mutant is unlikely to be folded and should not be able to interact with any of its known binding partners. This putative dominant-negative effect is hard to explain. The authors should inject an unrelated cRNA (e.g. GFP) into these oocytes and use those as the negative control. It is possible that the injection process per se perturbed the meiotic progression of these oocytes.

2) For the experiments in mouse oocytes, the authors should test mechanistically whether the checkpoint is turned off, for example, they could examine the localization of one of the checkpoint proteins by immunofluorescence.

3) As pointed out by reviewers #1 and #2, the references are often not correct and some important work in this area is not cited. Please ensure that all the relevant literature is cited in all of the appropriate places.

4) The authors should discuss the male phenotype more comprehensively.

5) Given the small sample size, the authors should make conclusions cautiously and ensure that statements take this caveat into account, particularly for the rescue experiments in human.

6) Although not essential, the addition of mechanistic work on the other two patient mutations in their mouse oocyte system would greatly strengthen the manuscript.

*Reviewer #1 (Recommendations for the authors):*

The study may be submitted to a more specialized journal. Rescue experiments of human oocytes need to be repeated, to reproduce the data.

*Reviewer #2 (Recommendations for the authors):*

Overall, I find this is an important translational study that adds to the growing body of literature that genetic mutations impact oocyte quality and fertility. However, to fully support your claims that the SAC is not being satisfied in patients who have a mutation in this gene, additional SAC assays in mouse oocytes should be performed to demonstrate this. I point you to papers by authors from Wassmann, FitzHarris, and Schindler labs for examples.

You also identified 3 variants, but only explore one. Is it possible that the other variants have different impacts on oocytes? This seems like a simple point to explore in your mouse oocyte system and would have an important clinical impact if therapeutic treatment would be different depending on the degree of truncation in the protein.

*Reviewer #3 (Recommendations for the authors):*

I only have one major point that needs to be addressed prior to publication. In Figure 3D, injection of the Mad2l1bp mutant cRNA into mouse oocytes delayed polar body extrusion as compared to the uninjected control. This suggested that the Mad2l1bp mutant had a dominant-negative function. On the other hand, this mutant is unlikely to be folded and should not be able to interact with any of its known binding partners. This putative dominant-negative effect is hard to explain. The authors should inject an unrelated cRNA (e.g. GFP) into these oocytes and use those as the negative control. It is possible that the injection process per se perturbed the meiotic progression of these oocytes.

---

## [Author Response]

Essential revisions:(1) In Figure 3D, injection of the Mad2l1bp mutant cRNA into mouse oocytes delayed polar body extrusion as compared to the un-injected control. This suggested that the Mad2l1bp mutant had a dominant-negative function. On the other hand, this mutant is unlikely to be folded and should not be able to interact with any of its known binding partners. This putative dominant-negative effect is hard to explain. The authors should inject an unrelated cRNA (e.g. GFP) into these oocytes and use those as the negative control. It is possible that the injection process per se perturbed the meiotic progression of these oocytes.

We have performed the GFP cRNA injection as suggested. See the reply in detail for Reviewer #3 below.

(2) For the experiments in mouse oocytes, the authors should test mechanistically whether the checkpoint is turned off, for example, they could examine the localization of one of the checkpoint proteins by immunofluorescence.

We have performed the SAC integrity check, as suggested, by immunofluorescence staining with Bub3. See the reply in detail for Reviewer #2 below.

(3) As pointed out by reviewers #1 and #2, the references are often not correct and some important work in this area is not cited. Please ensure that all the relevant literature is cited in all of the appropriate places.

As suggested, we have thoroughly checked and add more relevant reference literatures.

(4) The authors should discuss the male phenotype more comprehensively.

We have added more discussion in the manuscript.

5) Given the small sample size, the authors should make conclusions cautiously and ensure that statements take this caveat into account, particularly for the rescue experiments in human.

We have modified and toned down the statement as suggested in the manuscript.

6) Although not essential, the addition of mechanistic work on the other two patient mutations in their mouse oocyte system would greatly strengthen the manuscript.

We have explored the mechanism in the arrested human oocytes by single-cell RNA sequencing (Page 11). These data are in support of an essential role of MAD2L1BP in driving oocyte meiotic division. For the other two patients’ mutations, because they are much shorter than p.R285* in the first patient, we reasoned it is unnecessary to explore further. See the reply in detail for Reviewer #2 below.

Reviewer #1 (Recommendations for the authors):The study may be submitted to a more specialized journal. Rescue experiments of human oocytes need to be repeated, to reproduce the data.

We have performed more experiments, which were incorporated in new Figure.3, new Figure 3—figure supplement 1, and new Figure 4—figure supplement 1. For the rescue experiments, as mentioned by Reviewer 2, it is difficult to carry out in human oocytes. Because we only have 10 frozen human oocytes (two were used for Smart-seq2, eight were used for rescue experiment), we currently do not have additional oocytes for further rescue assay.

Reviewer #2 (Recommendations for the authors):Overall, I find this is an important translational study that adds to the growing body of literature that genetic mutations impact oocyte quality and fertility. However, to fully support your claims that the SAC is not being satisfied in patients who have a mutation in this gene, additional SAC assays in mouse oocytes should be performed to demonstrate this. I point you to papers by authors from Wassmann, FitzHarris, and Schindler labs for examples.

Thanks for the good comment. We have referred to the methods as you mentioned published from Wassmann, FitzHarris, and Schindler labs, and conducted the microinjection and immunofluorescence staining using one of the spindle assembly checkpoint components – Bub3. In support of our previous findings, we observed consistent activation of SAC machinery all through in vitro *maturation* upon the *Mut Mad2l1bp* cRNA injection, which indicates the intact SAC signaling. Consequently, these oocytes failed to undergo complete meiotic division I and possess bivalent chromosomes even after release for 16 hr. By contrast, the majority of oocytes with WT *Mad2l1bp* cRNA injection successfully undergo meiosis I and achieved typical univalent chromosomes with low levels of Bub3 staining (Figure 3—figure supplement 1). This additional SAC assay unambiguously confirms our previous results.

The following papers published from Wassmann, FitzHarris, and Schindler labs were also cited:

1. Karasu ME, Bouftas N, Keeney S, Wassmann K. Cyclin B3 promotes anaphase I onset in oocyte meiosis. J Cell Biol 218, 1265-1281 (2019).

2. Stein P, Schindler K. Mouse oocyte microinjection, maturation and ploidy assessment. J Vis Exp, (2011).

3. Aboelenain M, Schindler K, Blengini CS. Evaluation of the Spindle Assembly Checkpoint Integrity in Mouse Oocytes. J Vis Exp, (2022).

4. Charalambous C, Webster A, Schuh M. Aneuploidy in mammalian oocytes and the impact of maternal ageing. Nat Rev Mol Cell Biol 24, 27-44 (2023).

5. Mihajlovic AI, Haverfield J, FitzHarris G. Distinct classes of lagging chromosome underpin age-related oocyte aneuploidy in mouse. Dev Cell 56, 2273-2283 e2273 (2021).

6. Mihajlovic AI, FitzHarris G. Segregating Chromosomes in the Mammalian Oocyte. Curr Biol 28, R895-R907 (2018).

7. Vallot A, et al. Tension-Induced Error Correction and Not Kinetochore Attachment Status Activates the SAC in an Aurora-B/C-Dependent Manner in Oocytes. Curr Biol 28, 130-139 e133 (2018).

8. Touati SA, et al. Mouse oocytes depend on BubR1 for proper chromosome segregation but not for prophase I arrest. Nat Commun 6, 6946 (2015).

You also identified 3 variants, but only explore one. Is it possible that the other variants have different impacts on oocytes? This seems like a simple point to explore in your mouse oocyte system and would have an important clinical impact if therapeutic treatment would be different depending on the degree of truncation in the protein.

Thanks for your suggestion. We have identified *MAD2L1BP variants* from three patients*,* including two homozygous variants (p.R285* and p.R181*) and biallelic *MAD2L1BP variants* p.F173Sfs4*/ c.21-94G>A (Figure 1 and Sup Figure 1). Among these variants, only p.R285* is able to produce the longest truncated MAD2L1BP protein (with only loss of 22 residues) (Sup Figure 1). Since we have validated and confirmed that loss of 22 amino acids clearly abolished its interaction of MAD2L1BP (p.R285*) with the kinetochore protein MAD2 (Figure 2), there is no chance to establish the interaction between other much shorter MAD2L1BP variants (p.R181*, p.F173Sfs4*/ c.21-94G>A) and MAD2 protein. Therefore, the primary defect responsible for the patients’ infertility must be ascribed to the malfunction of MAD2L1BP-MAD2 interaction. For clarity, we have rephrased the texts in the manuscript.

Reviewer #3 (Recommendations for the authors):I only have one major point that needs to be addressed prior to publication. In Figure 3D, injection of the Mad2l1bp mutant cRNA into mouse oocytes delayed polar body extrusion as compared to the uninjected control. This suggested that the Mad2l1bp mutant had a dominant-negative function. On the other hand, this mutant is unlikely to be folded and should not be able to interact with any of its known binding partners. This putative dominant-negative effect is hard to explain. The authors should inject an unrelated cRNA (e.g. GFP) into these oocytes and use those as the negative control. It is possible that the injection process per se perturbed the meiotic progression of these oocytes.

Thanks for your good suggestion. Following your advice, we have injected the GFP cRNA as a mock control in both new Figure 3C and 3D (serving as a negative control). Consistently, we show GFP mock injection behaves phenotypically similar to non-injected group, suggesting that GFP injection per se has no adverse effect on normal meiotic progression, and somehow implicates a dominant-negative effect for Mad2l1bp Mut as compared with Mad2l1bp WT. Given that the cellular milieu of meiotic oocytes reported (less stringency of SAC machinery, large cytoplasmic volume) is in sharp distinct from that in somatic cells, we reasoned that there likely exists unknown partners in oocyte context, but not in soma, that could be affected by Mad2l1bp Mut. This disparity was indeed observed for kinetochore proteins (e.g., BubR1) between meiosis in oocytes and mitosis in soma (Refs: Homer, H., Gui, L. & Carroll, J. A spindle assembly checkpoint protein functions in prophase I arrest and prometaphase progression. Science. Touati, S. A. et al. Mouse oocytes depend on BubR1 for proper chromosome segregation but not prophase I arrest. Nat. Commun.). For clarity, we have rephrased the texts in the MS. This would be an interesting direction to explore the SAC responsive partners, e.g., through very sensitive mass-spectrometry-based proteomic approach, in varied cellular contexts in the future.